# Interplay between Autophagy and Herpes Simplex Virus Type 1: ICP34.5, One of the Main Actors

**DOI:** 10.3390/ijms232113643

**Published:** 2022-11-07

**Authors:** Inés Ripa, Sabina Andreu, José Antonio López-Guerrero, Raquel Bello-Morales

**Affiliations:** 1Departamento de Biología Molecular, Edificio de Biología, Universidad Autónoma de Madrid, Darwin 2, Cantoblanco, 28049 Madrid, Spain; 2Centro de Biología Molecular Severo Ochoa, Consejo Superior de Investigaciones Científicas (CSIC), Cantoblanco, 28049 Madrid, Spain

**Keywords:** ICP34.5, autophagy, herpes simplex virus type 1, neurovirulence

## Abstract

Herpes simplex virus type 1 (HSV-1) is a neurotropic virus that occasionally may spread to the central nervous system (CNS), being the most common cause of sporadic encephalitis. One of the main neurovirulence factors of HSV-1 is the protein ICP34.5, which although it initially seems to be relevant only in neuronal infections, it can also promote viral replication in non-neuronal cells. New ICP34.5 functions have been discovered during recent years, and some of them have been questioned. This review describes the mechanisms of ICP34.5 to control cellular antiviral responses and debates its most controversial functions. One of the most discussed roles of ICP34.5 is autophagy inhibition. Although autophagy is considered a defense mechanism against viral infections, current evidence suggests that this antiviral function is only one side of the coin. Different types of autophagic pathways interact with HSV-1 impairing or enhancing the infection, and both the virus and the host cell modulate these pathways to tip the scales in its favor. In this review, we summarize the recent progress on the interplay between autophagy and HSV-1, focusing on the intricate role of ICP34.5 in the modulation of this pathway to fight the battle against cellular defenses.

## 1. Introduction

Autophagy [1] is a highly conserved catabolic process among eukaryotes, consisting of the degradation of intracellular components into lysosomes to ensure metabolic homeostasis [2,3]. There are different types of autophagic pathways: macroautophagy, microautophagy, endosomal microautophagy and chaperone-mediated autophagy (CMA) [4]. The most studied pathway is macroautophagy, which involves the de novo generation of double-membrane vesicles called autophagosomes, which are subsequently fused with lysosomes for cargo degradation. Autophagosome formation is a highly regulated process carried out by functional units constituted by autophagy-related (ATG) proteins [5]. This process has been extensively reviewed in the recent years [6,7,8,9,10] and consists of five sequential steps, which include initiation, nucleation, elongation, closure, and fusion (Figure 1). 

First, an isolation membrane, called a phagophore, is created at the autophagosome formation site. The primary membrane source of phagophores is the endoplasmic reticulum (ER) [11], but the mitochondria [12], cell membrane [13], lipid drops [14], ER-Golgi intermediate compartment (ERGIC) [15] and recycling endosomes [16] have also been suggested as possible membrane sources. The first functional unit that takes part in the initiation of the phagophore is the Unc-51-like kinase (ULK) complex, formed by four components (ULK1/2, ATG13, FIP200/RB1CC1, and ATG101). The ULK complex targets membrane structures at the phagophore formation site and triggers the nucleation of the phagophore by the recruitment of a second autophagy functional unit, the class III phosphatidylinositol 3-kinase complex I (PI3KC3-C1), which contains the lipid kinase VPS34 together with accessory proteins (p150, Beclin-1, ATG14 and NRFB2). The cooperation of these components increases the local concentration of phosphatidylinositol-3-phosphate (PI3P), promoting the recruitment of PIP3-binding autophagy proteins to the phagophore membrane. For the initiation and nucleation steps, the role of the multi-spanning membrane protein ATG9A is also essential, which is located in vesicles and/or tubules in close proximity to phagophores. Through the transient association of ATG9A with phagophore membranes, the protein supplies key elements to the phagophore [17,18].

The following elongation and maturation of autophagosomal membranes involves two ubiquitin-like conjugations steps: the conjugation of ATG12 to ATG5 (with the subsequent recruitment of ATG16L1 to form a functional complex) and the post-translational modification of the microtubule-associated protein 1 light chain 3 (LC3), which belongs to the ATG8 family of proteins. First, ATG4 cleaves the C-terminal residues of LC3 to expose a glycine residue (LC3-I). Then, LC3-I is activated by the E1-like enzyme ATG7 and is transferred to the E2-like enzyme ATG3. The ATG12-5-16L1 complex, which functions as an E3-like enzyme, facilitates the transfer of LC3-I from ATG3 to a phosphatidylethanolamine (PE) of the autophagosomal membrane [19]. PE-conjugated LC3B-I, namely LC3B-II, is the most common marker used to monitor autophagy. The maturation and enclosure of autophagosomes are followed by the fusion of the outer autophagosomal membrane with a lysosome [20]. Finally, the inner autophagosomal membrane is degraded by lysosomal hydrolases, leading to the formation of a single-membrane mature autolysosome.

**Figure 1 ijms-23-13643-f001:**
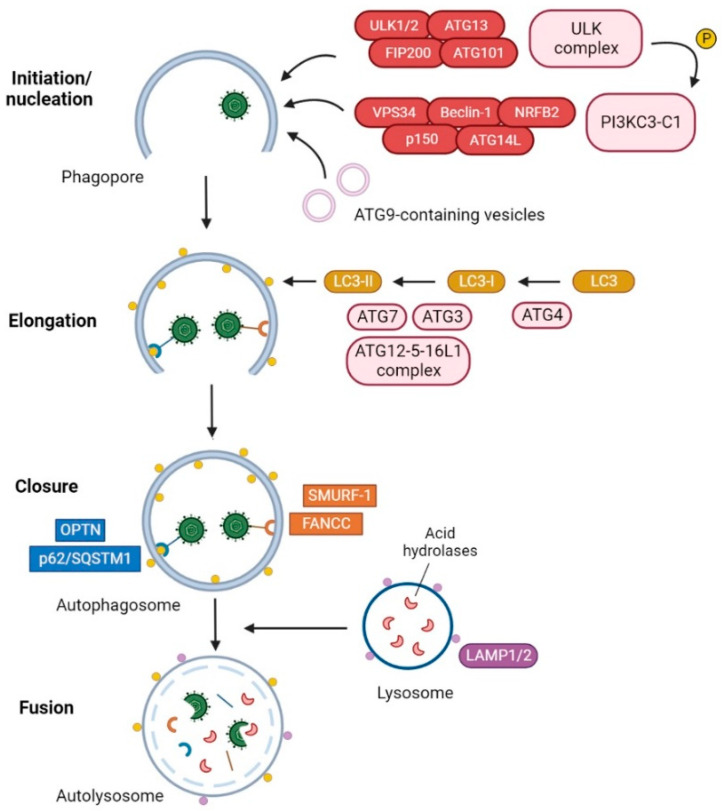
Degradation of HSV-1 by selective autophagy. The ULK complex initiates the formation of the phagophore by phosphorylating components of the PI3KC3 complex I and mediating the trafficking of ATG9-containing vesicles. The LC3-PE conjugation system participates in the elongation and closure of the double membrane, resulting in the formation of the autophagosome. The selective autophagy receptors (SARs) p62/SQSTM1 [21] and optineurin (OPTN) [22] and the autophagy receptor-like factors Fanconi anemia group C protein (FANCC) [23] and SMAD ubiquitin regulatory factor 1 (SMURF-1) [21] can interact with HSV-1 and mediate the recruitment of HSV-1 virions and/or viral cytoplasmic components into the autophagosomes. Once cargo has been engulfed, the external membrane of the autophagosome fuses with a lysosome for degradation of viral components.

Autophagy can be classified into non-selective or selective types depending on the cargo. In non-selective autophagy, bulk portions of the cytoplasm are sequestered by phagophores. This type of autophagy is usually stimulated under starvation conditions, due to its contribution to the maintenance of nutrient levels in cells. On the other hand, selective autophagy is based on the degradation of specific intracellular cargo through the activity of selective autophagy receptors (SARs). SARs interact with ATG8 family proteins on the inner membrane surface of phagophores, mediating the delivery of specific cargo and facilitating the recruitment of autophagic machinery [24,25]. In addition to maintaining the number and integrity of cellular organelles [26], selective autophagy also participates in pathogen clearance. The kind of selective autophagy involved in the sequestration and degradation of cytoplasmic viral components and/or entire virions is known as virophagy (Figure 1) [27]. This type of autophagy also contributes to the viral antigen processing and subsequent presentation on major histocompatibility complex (MHC) class I and II, which initiates adaptative immunity [28].

Many viruses can reverse and exploit multiple steps of autophagy, thus evading the immune response and facilitating viral replication [29]. This is the case of herpes simplex virus type 1 (HSV-1), which has acquired the ability to down-regulate the autophagic pathway. HSV-1 is a neurotropic virus that, after a primary infection in epithelial cells, traffics from axon terminals to the trigeminal ganglia and establishes latency [30]. Periodically, the virus can reactivate, leading to the most common symptoms of HSV-1 infection the painful blisters or open sores in or around the mouth. Occasionally, the virus may spread from the trigeminal ganglia to the brainstem, reaching the central nervous system (CNS) and causing herpes simplex encephalitis (HSE) [31]. This virus has also been proposed as an etiological agent or as a risk factor for neurological pathologies, such as Alzheimer’s disease [32] and multiple sclerosis [33].

Due to the high adaptability of the herpesviruses to their hosts, HSV-1 has developed several mechanisms to evade cellular defenses [34]. In this review, we will focus on the virulence factor ICP34.5, a viral multifunctional protein that blocks many facets of the cellular antiviral responses and plays an essential role in productive HSV-1 infection in several cell types [35], with the inhibition of autophagy as one of its most studied and controversial functions.

## 2. The HSV-1 Factor of Virulence ICP34.5

### 2.1. The HSV-1 γ34.5 Gene and the Infected Cell Protein 34.5

The infected-cell protein 34.5 (ICP34.5) of HSV-1 is encoded by the γ34.5 gene, also known as RL1. This gene is located in the terminal repeat (TR) sequences of the HSV-1 DNA, being present in two copies per genome [36]. The γ34.5 gene is classified as a gamma-1 or leaky late gene. Although maximal levels of ICP34.5 are reached in the late phase of infection, the expression and function of the protein at early times, 3 hours post-infection, are essential for the control of the host cell environment and for successful viral replication [37]. The γ34.5 gene lacks a canonical TATAA box. Part of its promoter and the transcription initiation site are within the unique repeat junction called the “a” sequence, and its 5′unstranscribed domain (UTR 5′) is GC rich and has several repeats, lacking the characteristic features of HSV-1 promoters [38].

The structure of ICP34.5 consists of a disordered N-terminal domain, followed by a linker region with a variable number of ATP (Ala-Thr-Pro) repeats, and a conserved C-terminal domain (Figure 2). The C-terminal domain of ICP34.5 is highly homologous to the C-terminal of two conserved mammalian proteins implied in cellular stress responses: the growth arrest DNA damage-inducible protein GADD34 [39] and the myeloid differentiation primary response protein MyD116 [40]. These proteins are structurally related. Their corresponding domains are functionally interchangeable with the C-terminal of ICP34.5 in infected cells, and it was proposed that, in the course of virus evolution, HSV-1 “borrowed” this gene domain to deal with cellular defenses [41].

ICP34.5 is a highly dynamic protein that shuttles between the nucleus, the nucleolus, and the cytoplasm, with its function dependent on cellular location [42]. Generally, during early times of HSV-1 infection, the protein is mainly located in the nucleus, and, at 8–12 h post-infection, it is accumulated in the cytoplasm [43]. The N-terminus of ICP34.5 contains an arginine-rich cluster, which primarily targets the protein of the nucleolus, and a leucine-rich cluster, which acts as a nuclear export signal. The C-terminus contains a bipartite nuclear localization signal, consisting of two clusters of basic amino acids separated by a 9–11 amino acids spacer, which is implied in the import of ICP34.5 into the nucleus [42].

The amino and carboxyl-terminal domains of ICP34.5 are mainly constant, but the number of ATP repeats of the linker region varies between 5 and 10 depending on the HSV-1 strain. Variations in the number of ATP repeats may indirectly affect the cellular location of the protein, likely by masking or unmasking nuclear signals. ICP34.5 with a higher number of ATP repeats tends to be in the cytoplasm [44]. Moreover, the number of ATP repeats determines not only the ICP34.5 cellular localization, but also the tissue-behavior and the virulence of the virus [45].

### 2.2. HSV-1 ICP34.5: A Catch-All

ICP34.5 is primarily described as a neurovirulence factor essential for HSV-1 infection of neurons in vivo. HSV-1 lacking the γ34.5 gene is profoundly neuroattenuated in mouse brains [36,46]. However, the role of ICP34.5 in viral pathogenesis is strongly dependent on both cell type and cell stage, and it is not only restricted to neuronal cells. For instance, ICP34.5 is required for the HSV-1 infection of stationary-state mouse embryo fibroblast 3T6 cells, whereas the replication of ICP34.5-deficient HSV-1 in hamster kidney fibroblasts BHK cells is indistinguishable from the wild-type virus [47]. The role of ICP34.5 in HSV-1 infection is extremely complex, and the reason for the strong cell type dependence is largely unknown. Even when ICP34.5 is considered an essential virulence factor for the HSV-1 infection of the CNS, ICP34.5-deficient viruses retain the ability to infect and destroy the mouse ependyma (the layer formed by epithelial glial cells that lines the surface of the ventricular system and the central canal of the spinal cord) [48]. In the following sections, we describe and discuss the diverse functions of ICP34.5 during HSV-1 infection, with these roles classified according to the protein domain that is involved.

#### 2.2.1. Functions of the Carboxyl Domain of HSV-1 ICP34.5

The first function of ICP34.5 to be described is associated with the C-terminal domain. Upon viral infection, host cells detect the presence of the virus through pattern recognition receptors (PRRs). PRRs recognize conserved viral structures called pathogen-associated molecular patterns (PAMPs), which include viral proteins, DNA and RNA [49]. PRRs relay signals to TANK-binding kinase 1 (TBK1), which phosphorylates the interferon (IFN) regulatory factor 3/7 (IRF3/7). Phosphorylated IRF3/7 is translocated to the nucleus to induce the production of type I IFN and other cytokines. In response to IFN, several molecules are activated through the JAK/STAT pathway, establishing an antiviral environment in the neighboring cells [50,51]. Among these antiviral molecules is dsRNA-dependent protein kinase (PKR), a cytoplasmic protein capable of detecting viral dsRNA. Its major role is the inhibition of both cellular and viral protein synthesis through the phosphorylation of the translation initiation factor 2 subunit α (elF2α) [52]. HSV-1 can block this important antiviral mechanism through ICP34.5 [53] (Figure 3). The C-terminal domain of ICP34.5 recruits elF2α and the protein phosphatase 1α (PP1α) [54,55]. PP1α dephosphorylates elF2α, inhibiting the PKR pathway and the translational arrest. Alternatively, ICP34.5 induces the translocation of the nucleolar protein NOP53 to the cytoplasm, which facilitates the recruitment of PP1α [56].

#### 2.2.2. Functions of the Amino Domain of HSV-1 ICP34.5

The functions of ICP34.5 associated with its N-terminal domain (Figure 3) remain obscure. On the one hand, the 68–87 amino acid region of ICP34.5 or Beclin-1-binding domain (BBD) recruits and blocks an essential protein for autophagosome biogenesis, Beclin-1, thus inhibiting autophagy [57]. Initially, the function of BBD was understood to be restricted to the prevention of autophagy, but an additional role of this domain was recently discovered. BBD not only interacts with Beclin-1, but also with multiple regulators of the antioxidant response, mitochondrial trafficking, and programmed cell death. In HSV-1-infected primary mouse neurons, the interaction of BBD with the sensor of oxidative stress KEAP1 and the regulator of mitochondrial dynamics PGAM5 causes the formation of mitochondrial clusters near the nucleus, where virions are produced. This mitochondrial distribution relieves the bioenergetic demands on the cell during infection and facilitates transcription under oxidative stress [58].

On the other hand, the N-terminal domain of ICP34.5 plays an important role in the fight against IFN responses by the dephosphorylation of IRF3. Initially, it was described that TBK1 interacts with ICP34.5, thereby inhibiting IRF3 phosphorylation. ICP34.5 was observed to bind to TBK1 in the 76–106 amino acid region of the N-terminus, known as the TBK-1-binding domain (TBD). HSV-1 lacking the γ34.5 gene replicates efficiently in TBK1(-/-) cells but not in TBK1(+/+) cells [59,60]. However, later studies demonstrated that TBD binds overexpressed but not endogenous TBK1. Furthermore, TBD shares a region (72–87 aa) with BBD (68–87 aa). Infection with a recombinant virus that lacks amino acids 87 to 106, a portion of the previously described TBD, which is outside the region where TBD and BBD overlap, shows no impact on IRF3 phosphorylation or on viral replication and pathogenesis in mice. IRF3 phosphorylation is independent of TBD, and the ability of ICP34.5 to control IRF3 activation was suggested to be a consequence of either the reverse translational shutoff or the expression of other viral IFN inhibitors [61].

The dephosphorylation of IRF3 by HSV-1 can also be mediated by the inactivation of the stimulator of interferon genes (STING), a key component of innate immunity. Upon HSV-1 infection, intracellular dsDNA sensors recognize the HSV-1 genome and activate STING, which recruits TBK1 to phosphorylate IRF3 and initiates the production of IFN [62]. However, the N-terminal region of ICP34.5 can bind to and inactivate STING, blocking the HSV-1 DNA sensing pathway and, consequently, the IFN response. In addition, the N-terminal domain of ICP34.5 disrupts the translocation of STING from the ER to the Golgi apparatus, a process necessary to promote cell-mediated immunity [63].

Finally, the N-terminus of ICP34.5 also mediates the reorganization of nuclear lamins [64]. The nuclear lamina is a meshwork in the inner nuclear membrane composed primarily of type V intermediate filament proteins, lamins A/C and B. This lamina acts as a barrier for the transit of viral nucleocapsids to the cytoplasm. However, herpesviruses can alter this lamina to promote viral nuclear egress [65]. HSV-1 ICP34.5 can interact with the protein kinase C (PKC) and the complement component 1 Q subcomponent-binding protein (C1QBP/p32) [66] through its N-terminal domain, and it redirects these proteins to the nuclear membrane, where the UL31/UL34 complex resides. The formation of this multiprotein complex activates PKC, which phosphorylates lamin A/C, promoting the disintegration of the nuclear lamina and HSV-1 nucleocapsid egress [67].

#### 2.2.3. Functions of the Amino and Carboxyl Domains of HSV-1 ICP34.5

Both amino and carboxyl terminal regions are required for certain roles of ICP34.5 (Figure 3). Among these functions is the suppression of dendritic cells (DCs) maturation. Unlike the wild-type virus, HSV-1 lacking the γ34.5 gene stimulates the expression of MHC-II, CD86 and cytokines in immature DCs [68]. The mechanism by which ICP34.5 disrupts the maturation of DCs is mediated by the dephosphorylation of IκB kinase. The N-terminal domain of ICP34.5 binds to the protein IKKα/β and the C-terminal domain recruits PP1α. This multiprotein complex dephosphorylates IκB kinase to prevent the activation of NF-κB, a transcription factor that regulates the expression of genes involved in immune response, inflammation, and cell survival [69].

In non-dividing cells, the full-length ICP34.5 can also be required to overcome the blocking of DNA replication. During the early phase of HSV-1 infection, ICP34.5 is mainly located in the nucleus, where it can interact with DNA. This interaction is mediated by the recruitment of HSV-1 DNA replication proteins, such as UL30 polymerase, and the proliferating cell nuclear antigen (PCNA), an essential protein for the replication and repair of cellular DNA. HSV-1 infection promotes PCNA translocation into the nucleus, where the protein interacts with ICP34.5 to promote the initiation of viral replication [43].

Recently, it was discovered that, although the N-terminus of ICP34.5 blocks the HSV-1 DNA-sensing pathway, the full-length protein is needed for the inhibition of the RNA-sensing pathway. The retinoic acid-inducible gene-I (RIG-I) protein is a receptor that recognizes cytoplasmic viral RNA. Upon activation, RIG-I interacts with the mitochondrial antiviral signaling protein (MAVS) and activates the production of IFN via IRF3 phosphorylation. ICP34.5 precludes the assembly of RIG-I with the cellular chaperone 14–3-3ε, preventing the translocation of RIG-I from the cytosol to the mitochondria. Cytosolic RIG-I is unable to interact with MAVS and the IFN response initiated through the recognition of viral RNA is blocked [70].

## 3. HSV-1 Modulation of Autophagy

### Induction of Autophagy at the Early Stages of HSV-1 Infection

Under stress conditions, such as nutrient starvation, oxidative stress, the presence of unfolded proteins or pathogen infections, autophagy can be stimulated to promote metabolic adaptation and cellular survival. As a result of the role of this pathway in cellular protection against pathogens, it is not surprising that, in early stages of HSV-1 infection, autophagy flux is usually stimulated (Figure 4). However, the molecular mechanisms involved in autophagy stimulation during HSV-1 infection are still not fully understood. The apparent cell-type dependence of these mechanisms makes their research strongly complicated.

In murine embryonic fibroblasts and sympathetic neurons, the IFN-inducible PKR signaling pathway not only leads to cellular translational arrest, but also to the induction of autophagy and HSV-1 degradation into autolysosomes. The stimulation of autophagy by this pathway appears to be dependent on HSV-1 gene expression [71,72,73]. Viral gene expression is also necessary for the induction of autophagy in monocytic-derived DCs, because infection with UV-inactivated HSV-1 does not lead to the stimulation of this degradative pathway [74]. The relevance of PKR mediating selective autophagy has been demonstrated not only in vitro, but also in vivo [57].

Emerging evidence has shown that toll-like receptors (TLRs) can also mediate the induction of autophagy. TLRs are receptors that recognize PAMPs and reside within the cell surface membrane and in endosomal compartments [75]. HSV-1 may be recognized by the surface TLR2 [76,77,78]. Upon activation, TLR2 can recruit the myeloid differentiation primary response protein MyD88, causing the dissociation of Beclin-1 from the B cell lymphoma-2 (BCL-2) inhibitory complex, which results in the induction of autophagy [79]. The MyD88 protein is necessary to stimulate autophagy in HSV-1-infected monocytic THP-1 cells [80].

The presence of the HSV-1 genome in the cytoplasm can also induce autophagy by the DNA-sensing pathway. The DNA sensor cyclic GMP–AMP (cGAMP) synthase cGAS, upon recognizing the HSV-1 genome, produces the second messenger cGAMP to initiate the STING pathway, leading to TBK1 activation and IFN production. The STING pathway has been proposed to play a role in the promotion of autophagy in HSV-1 infected myeloid cells [81] and fibroblasts [82]. Although STING-autophagy induction is dependent on TBK1, due to the significant impairment of TBK1-deficient fibroblasts in autophagy stimulation, this pathway seems to be IFN-independent. TICAM1/TRIF null mutant fibroblasts, which are deficient in IFN production, can induce autophagy [83,84].

## 4. Selective Inhibition of Autophagy by HSV-1 ICP34.5

Upon the initial induction of autophagy, HSV-1 may counteract host cells by down-modulating the pathway through ICP34.5. The role of ICP34.5 in autophagy inhibition is restricted to a specific domain (amino acids 68 to 87) implicated in the recruitment of Beclin-1 [57], an essential protein for autophagosome formation [85]. This domain is known as “Beclin-1-binding domain” or BBD. The C-terminal region of the protein is dispensable for this autophagy-inhibitory activity, suggesting that the modulation of autophagy is independent of PP1α/elF2α binding to ICP34.5 [57]. However, this is not the end of the story…

The effect of HSV-1 ICP34.5 on autophagy is strongly dependent on the cell type. Although BBD inhibits the formation of autophagosomes in neurons and fibroblasts [73], in bone-marrow DCs (BM-DCs) it prevents autophagosome maturation, causing the accumulation of long-lived proteins and autophagosomes [86]. In some cell types, including corneal epithelial and retinal ganglion cells, the protein has no effect on cellular autophagy [87]. Finally, it is important to make a distinction between the inhibition of autophagy, based on the down-modulation of basal autophagy, and the inhibition of its stimulation in response to infection. In monocyte-derived DCs [74] and in BM-DCs, ICP34.5 does not inhibit the induction of autophagy during early times of infection. In BM-DCs, the HSV-1 DNA-sensing pathway induces autophagy in a PKR-independent manner that cannot be antagonized by ICP34.5 [81].

Autophagy suppression by the virus not only prevents virion removal into autolysosomes, but it also prevents the processing and delivery of intracellular viral antigens to MHC class I and II molecules, inhibiting adaptative immunity [88] (Figure 4). When murine DCs are infected with ICP34.5-deficient HSV-1, viral antigen presentation on MHC-I is increased compared to the wild-type virus [89]. In mice, HSV-1 lacking BBD precludes autophagy-mediated MHC-II antigen presentation, decreasing the stimulation of CD4+ T cells [90].

The reduced presentation of viral antigens, a consequence of HSV-1 autophagy arrest, may be compensated by host cells by an alternative Beclin-1-independent autophagic pathway. Recently, the presence of four-layered membrane structures, containing the autophagy marker LC3B, was described in HSV-1 infected macrophages. These four-layered structures are formed by rolling up the inner and outer nuclear membrane, and they are accumulated in the cytoplasm about eight hours post-infection. These vacuole structures enhance the presentation of endogenous viral antigens on MHC-I molecules, providing an additional pathway apart from viral antigen degradation by the proteasome. Whereas the formation of canonical autophagosomes can be promoted by pharmacological autophagy inducers, the generation of these four-layered structures is only induced by HSV-1 infection, which means that it is regulated in a way that is different from macroautophagy [88]. This autophagic pathway was denominated as NEDA (Nuclear-Envelope Derived Autophagy) (Figure 4). The first evidence of this process was notified in macrophages, but it has later been observed in many different cell types, indicating that it is a general host mechanism against HSV-1 infection. NEDA is regulated by ICP34.5 but is Beclin-1 and BBD independent. Interestingly, the inhibition of host translation shutoff through the C-terminal domain of ICP34.5 is required for NEDA stimulation [91]. Furthermore, although the activation of TBK1 by the DNA-sensing pathway mediates the induction of macroautophagy in HSV-1 infected cells, NEDA is a TBK1-independent pathway suggested to be a general cellular response to stress [83]. These results demonstrate the relevance of analyzing macroautophagy and NEDA separately. Whereas LC3B is a common marker for both processes, LC3A is only present on NEDA structures, and it can be used to disguise both types of autophagy [91].

To deal with virophagy, HSV-1 suppresses macroautophagy by blocking Beclin-1 through ICP34.5. Besides, HSV-1 prevents the PKR translational arrest by the recruitment of elF2α and PP1α in the C-terminus of ICP34.5. As a general stress response, the host cell initiates an alternative autophagy pathway known as NEDA. This process consists of the Beclin-1-independent formation of four-layered membrane structures by the coiling of the nuclear membrane. NEDA is suggested to have an antiviral role promoting viral antigen presentation, but further research about the cellular functions of NEDA is required.

## 5. The Intricate Role of the Beclin-1-Binding Domain of ICP34.5 in HSV-1 Virulence

HSV-1 strains lacking the BBD of ICP34.5 exhibit reduced neurovirulence and viral replication in mice compared to wild-type viruses [57,61,90]. However, whereas HSV-1 lacking BBD is strongly neuroattenuated in the brains of mice, this protein domain is dispensable for productive HSV-1 replication in the neuronal established cell line SK-N-SH [57]. Even in primary fibroblast cultures, no difference is noticeable in viral replication between HSV-1 lacking BBD and the wild-type virus [92].

HSV-1 lacking BBD is not as attenuated in vitro as the data would predict. Recently, it has been proposed that the differences in virulence between in vivo and in vitro studies could be due to the presence of ferric nitrate in culture media. In the presence of this iron salt, which is involved in redox reactions, the attenuated replication of HSV-1 lacking BBD in primary human fibroblasts is partially restored. The cause of the masked BBD effect can be explained by the fact that this domain recruits not only Beclin-1, but also multiple mitochondrial regulators that play a relevant role in redox reactions and cellular metabolism [58]. Unfortunately, primary neuron cultures do not survive in redox buffer-free medium conditions. Hence, viral replication experiments cannot be performed under these conditions, where the effect of BBD is not masked.

Although the role of BBD in virulence may not be restricted to the inhibition of Beclin-1-autophagy, this does not mean that the suppression of autophagic flux by BBD has no effect on infection. Although BBD-deleted HSV-1 is significantly neuroattenuated in mice, the infectivity of this virus is restored in pkr -/- mice, which have an impairment in PKR-autophagy induction. These results demonstrate that the inhibition of PKR autophagy by BBD is an important HSV-1 mechanism of neurovirulence in vivo [57].

## 6. ICP34.5 Is Not Alone in HSV-1 Autophagy Inhibition

HSV-1 exploits the factor of virulence ICP34.5 to inhibit Beclin-1-dependent autophagy. However, ICP34.5 is neither the unique HSV-1 protein capable of modulating autophagy, nor is Beclin-1 the only target used by the virus for autophagy suppression (Figure 5).

Us11 is an HSV-1 late protein that binds to dsRNA and physically interacts with PKR [93]. Through the direct association to PKR, Us11 prevents the activation of the PKR/elF2α signaling pathway and, consequently, translational arrest [94,95]. Although Us11 is not able to interact with Beclin-1, its interaction with PKR has a strong anti-autophagic activity with cells. Thus, autophagy inhibition in HSV-1 infected cells can be a result of the activity not only from ICP34.5, but also from Us11 [96]. Us11 also reduces the formation of autophagosomes by the disassembly of the functional complex formed by TRIM23 (tripartite motif protein 23), Hsp90 (heat shock protein 90) and TBK1 [97]. Tripartite motif (TRIM) proteins are essential regulators of both the IFN-response and autophagy [98], and current evidence suggests that TBK1 activation through TRIM23 is a key step in selective autophagy in multiple viral infections, including HSV-1 [99]. Nonetheless, the precise mechanism involved in the inhibition of TBK1-selective autophagy by Us11 remains to be fully understood.

In addition to preventing autophagosome formation, recent research suggests that viruses can evade virophagy by targeting SARs [100]. HSV-1 may down-modulate two important SARs implied in the recognition and delivery of specific viral ubiquitinated cargo to the phagophores: the autophagy receptor sequestosome 1 (p62/SQSTM1) and the mitophagy adaptor optineurin (OPTN). Cellular levels of p62/SQSTM1 and OPTN are significantly reduced after 3–6 h of HSV-1 infection in various cell lines [101]. After activation by TRIM23, TBK1 phosphorylates and activates p62/SQSTM1 and OPTN [102]. However, it has not been established if there is a relationship between the inhibitory effect of Us11 on the TRIM23-TBK1 complex and the early down-modulation of these SARs. In contrast, it has been observed that, upon HSV-1 infection, both SARs are degraded in the proteasome by a mechanism that requires ICP0 expression and calcium mobilization, but the role of calcium in ICP0 activity is currently unknown [101].

The involvement of ICP34.5 in p62/SQSTM1 and OPTN modulation has been also analyzed. In cells infected with ICP34.5-deficient HSV-1, these proteins remain down-regulated, indicating that ICP34.5 is not involved in this process. However, ICP34.5-deficient HSV-1 is unable to inhibit macroautophagy, and SARs can be degraded into autolysosomes [101]. On the other hand, cells infected with HSV-1 lacking the γ34.5 gene show a reduced accumulation of ICP0, so it is possible that ICP34.5 modulates the cellular levels of p62/SQSTM1 and OPTN in an indirect manner [61].

To determine the effect of these SARs on HSV-1 infection, p62/SQSTM1 and OPTN knockdown cells were infected. No viral growth defect was observed, possibly because HSV-1 usually down-modulates both proteins early after infection. However, the exogenous expression of p62/SQSTM1 negatively affected viral yields [101]. It remains undetermined if SARs down-regulation by HSV-1 is advantageous for infection because of the role of these receptors in autophagy, or due to other functions they perform in cells. p62/SQSTM1 also has roles in cellular metabolism, cell signaling and apoptosis [103], and OPTN is implied in mitophagy, a type of selective autophagy involved in the degradation of damaged mitochondrial components, and modulates multiple cellular processes related to protein trafficking and membrane cargo delivery from the Golgi apparatus to the plasma membrane [104].

Recently, it was discovered that the HSV-1 protein Ser/Thr kinase Us3 can also suppress autophagy [105]. Us3 may phosphorylate and activate the nutrient-sensing mammalian target of rapamycin kinase complex 1 (mTORC1) [106], which is a negative regulator of the ULK complex that keeps autophagy inactive under physiological conditions [107]. In addition, Us3 can modulate autophagy downstream of mTORC1, by the phosphorylation and inactivation of both the ULK1 complex and Beclin-1 [105].

HSV-1 has developed multiple mechanisms to prevent autophagy. However, the question remains regarding why the virus requires different strategies to inhibit the same pathway. Two options have been considered. On the one hand, HSV-1 can antagonize autophagy in a distinct manner in neuronal and non-neuronal cells. In non-neuronal cells, Us3 seems to have a main role in autophagy inhibition, predominant over the action of ICP34.5. Although the replication of ICP34.5 null mutant HSV-1 in fibroblasts is not rescued by suppressing autophagy, the replication of Us3-deficient and Us3-ICP34.5 doubly deficient HSV-1 is partially restored [105]. On the other hand, HSV-1 may inhibit not only Beclin-1-dependent autophagy, but also Beclin-1-independent or non-canonical autophagic pathways by using other virulence mechanisms [108].

## 7. The Two Sides of Autophagy

Macroautophagy is broadly considered an innate immune response that protect cells from viral infections [109]. Due to the antiviral role of autophagy, pharmacological modulation of this pathway has been proposed as a potential tool to fight HSV-1 infection [110]. Autophagy stimulation by physiological (cellular starvation in a minimal medium) and pharmacological approaches (cellular treatment with autophagy inducers such as MG-132 or trehalose) significantly suppresses HSV-1 infection in neuronal and non-neuronal cell types [101,111]. However, pharmacological autophagy inducers perform multiple cellular functions, which may also influence HSV-1 replication. For instance, MG-132 is a proteasome inhibitor that is not only associated with the induction of autophagy, but also with ER stress and apoptosis [112]. Thus, decreased infection may be a consequence not of autophagy stimulation, but of the unspecific effects of the drug. When neuroblastoma cells are transfected with Beclin-1-expressing plasmids to stimulate autophagy, instead of using pharmacological inducers, the hyperactivation of the pathway may only slow the rate of HSV-1 replication [113]. These results suggest that the main effect of pharmacological autophagy inducers on HSV-1 infection may not be a consequence of autophagy stimulation. Further research about the mechanisms of action of these drugs in the impairment of HSV-1 infection is needed.

Although the antiviral role of autophagy in neurons has been demonstrated in vitro and in vivo, in epithelial and other mitotic cells this pathway appears not to be a predominant defense mechanism. Keratinocytes unable to undergo autophagy were generated from Atg5-KO mice, with ATG5 being an essential protein for autophagosome formation. These keratinocytes showed no differences in HSV-1 replication compared to those from wild-type mice. The antiviral contribution of autophagy could be masked in wild-type mice by the inhibition of the pathway through ICP34.5. To rule out this possibility, wild-type mice were infected with BBD-deleted HSV-1, but viral replication was indistinguishable from the wild-type virus [114]. Similar results were observed in the MEFs of Atg5-KO mice [92]. However, it has recently been proposed that, actually, autophagy does have an important antiviral role in these cells, but the virus suppresses the pathway mainly with Us3, not ICP34.5 [105]. The antiviral potential of autophagy in non-neuronal cells might be underestimated, since previous studies have focused on the role of ICP34.5 and were carried out with HSV-1 expressing Us3, a protein which would inhibits PKR- and Beclin-1-dependent autophagy, likely in a stronger manner that of ICP34.5.

Autophagy is often considered a double-edged sword, and a possible proviral role of this pathway in HSV-1 infection was previously raised [115]. The suppression of autophagy, both by pharmacological inhibitors and by siRNA knockdown technology, significantly impairs HSV-1 infection in human acute monocytic leukemia (THP-1) cells and in primary human monocytes. These results indicate that the virus may use the autophagic machinery for its own benefit [80]. One proposed proviral function of autophagy is the degradation of the nuclear lamina [74]. The autophagy protein LC3 is present not only in the cytoplasm, but also in the nucleus, where it can interact directly with the lamins. This interaction can lead to the transport of the nuclear lamins from the nucleus to the cytoplasm for lysosomal degradation [116]. Lamina disintegration by nuclear autophagy has been proposed to facilitate the egress of HSV-1 capsids from the nucleus. This kind of autophagy is not down-modulated by ICP34.5, which means that, similar to NEDA, it is Beclin-1 independent [74]. 

Finally, autophagy may participate in the formation of microvesicles (MVs) during HSV-1 infection [117]. MVs are extracellular vesicles formed by direct outward budding, or pinching, of the plasma membrane. These vesicles may play an important role in viral infections, promoting viral dissemination and evasion of the immune system [118,119]. HSV-1 infected cells show greatly increased production of MVs compared to mock-infected cells, and non-enveloped HSV-1 virions have been observed inside LC3B-positive MVs. These MVs appear to mediate viral dissemination and cellular tropism, and they contribute to avoiding immune surveillance. However, further research is required to describe the specific role of autophagosomal membranes in the formation of MVs during HSV-1 infection [120,121].

Most of the research is focused on the effect of autophagy stimulation or inhibition in HSV-1 infection. However, the knowledge about the influence of basal autophagy on HSV-1 replication is scarce. The level of basal autophagy is strongly dependent on cell type, and it has been proposed as an important host-range determinant. In contrast to immature DCs, mature DCs are non-permissive for HSV-1 infection, which seems to be a consequence of its inefficient autophagic flux. In these cells, nuclear lamina degradation is impaired and fewer viral capsids are released from the nucleus to the cytoplasm [74]. However, very high levels of autophagy also appear to impair HSV-1 infection. Fibroblasts with a higher level of basal autophagy are better protected against HSV-1 infections [122]. Canonical and non-canonical autophagic pathways are highly regulated processes involved in numerous cellular functions, and a fine-tuning of these pathways could be essential for virus replication. In ocular cells, both the suppression and induction of autophagy significantly diminish HSV-1 infection. Because autophagy may function as a dual-edged sword, maintaining an intermediate level of autophagy may be required for a successful HSV-1 infection [123].

## 8. Herpes Simplex Virus Type 2 and Autophagy

Herpes simplex virus type 2 (HSV-2) is an alphaherpesvirus that primarily infects genital epithelial cells and establishes latency in sensory neurons of the sacral dorsal root ganglia. HSV-2 infection can cause severe genital diseases, with this virus being the most common cause of genital ulcers [124]. Although the modulation of autophagy by HSV-1 has been extensively analyzed, research focused on HSV-2 is scarce. It has been observed that HSV-2 inhibits autophagy in genital epithelial cells and the treatment with an autophagy inducer, known as JZ-1, significantly impairs viral infection in vitro [125] and in vivo [126]. Similar to HSV-1, the modulation of autophagy by HSV-2 seems to be cell-type dependent [47], and basal autophagy appears to play an important role for HSV-2 productive infection [127].

However, although HSV-2 can inhibit the autophagic flux similarly to HSV-1, the anti-autophagic proteins that are involved in preventing autophagy have not yet been determined. ICP34.5 is an important virulence factor for HSV-2, since deletion of the γ34.5 gene leads to a severe reduction in infection [128,129], but it remains unknown if HSV-2 down-modulates autophagy through ICP34.5. The amino acid sequence of the N-terminus of HSV-1 ICP34.5 is only 41% identical to the first exon of HSV-2 ICP34.5, and insertions appear to disrupt the corresponding HSV-1 BBD [129].

There are still more differences between HSV-1 and HSV-2 ICP34.5 apart from their amino acid sequences. In contrast to HSV-1, the γ34.5 gene of HSV-2 has been predicted to contain a 154 pb intron. This intron seems not to affect the function of the protein [130]. In addition to the full-length protein (ICP34.5α), a shorter form (ICP34.5β) is generated by alternative splicing of the HSV-2 γ34.5 gene. This novel splice-variant lacks the C-terminal conserved GADD34 domain and is unable to inhibit PKR-mediated elF2α phosphorylation. Although both forms of ICP34.5 are detected in the nucleolus, ICP34.5α is mainly located in the cytoplasm and ICP34.5β is in the nucleus. The cellular function of the ICP34.5 splice-variant has not been established [129]. Furthermore, two other forms of ICP34.5 have been identified. These novel forms of ICP34.5 lack the N terminus and are not products of alternative splicing or internal transcript initiation. The diversity of proteins generated from the HSV-2 γ34.5 gene implies additional levels of regulation through potential independent functions associated with the different forms of ICP34.5 [131].

Finally, it was discovered that the expression of the HSV-2 γ34.5 gene can be controlled by microRNAs related to the latency-associated transcript (LAT). Authors have hypothesized that these microRNAs may modulate the outcome of viral infection in the peripheral nervous system by reducing the expression of ICP34.5 [132,133]. The silencing of ICP34.5 by microRNAs has not been reported in HSV-1 infections.

## 9. Conclusions

Autophagy is a fine-tuned and regulated process, and, when “autophagic balance” tips to one side, HSV-1 infection is often impaired. As a result of the strong influence of this pathway in viral infections, the modulation of autophagy has been suggested as a potential future therapy against HSV-1. However, before becoming fully involved in this type of treatments, it should be considered that unbalanced autophagy can negatively affect not only viral infection, but also cellular homeostasis and survival.

Autophagy is especially important in maintaining the correct functionality of the CNS. Deregulation of this pathway, whether excessive or insufficient, has been related to severe neurodegenerative disorders. Moreover, the suppression of autophagy by HSV-1 in the CNS has been proposed as a cause of neurodegeneration. The “autophagic balance” is extremely sensitive, and further research is required to decode the mechanisms that allow us to modulate autophagy accurately.

## Figures and Tables

**Figure 2 ijms-23-13643-f002:**
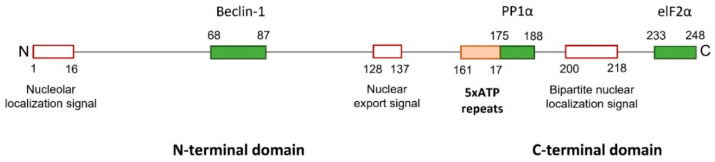
ICP34.5 of HSV-1 strain 17. HSV-1 ICP34.5 is a protein with 248 amino acids that can be divided into three regions: the amino and the carboxyl terminal domains, and a linked tandem of five ATP repeats (161–175 aa) (orange box). The N-terminal region of ICP34.5 contains a nucleolar localization and nuclear export signals, and the C-terminal domain includes a nuclear localization signal (red boxes). Three binding domains (green boxes) have been characterized: the Beclin-1-binding domain (BBD) in the N-terminus, and the PP1α- and elF2α-binding domains in the C-terminus.

**Figure 3 ijms-23-13643-f003:**
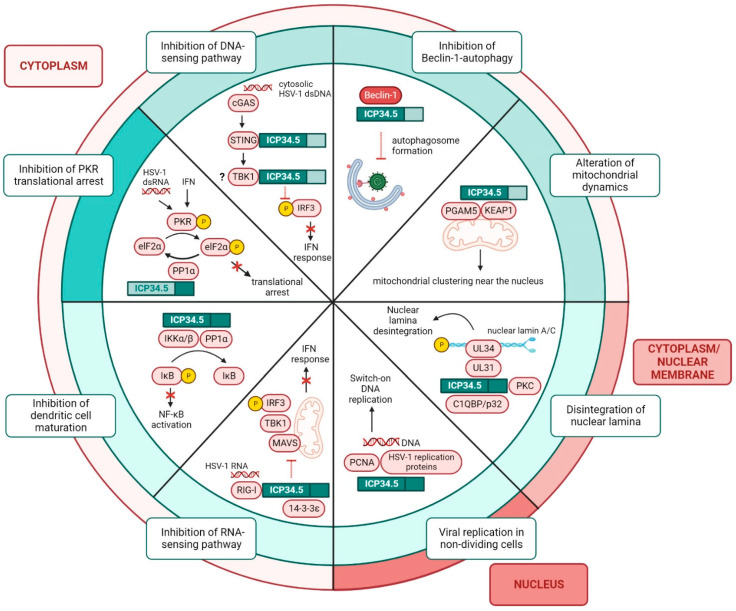
Functions of ICP34.5 in HSV-1 infected cells. Functions of ICP34.5 can be classified according to the cellular location of the protein, which shuttles between the cytoplasm and the nucleus. The amino and the carboxyl domains of ICP34.5 play different roles in infected cells. The C-terminus prevents the translational arrest by the binding of PP1α and the subsequent dephosphorylation of elF2α. The N-terminus is involved in the suppression of Beclin-1-autophagy, the degradation of the nuclear lamina to facilitate nucleocapsid egress from the nucleus, and the blockade of IFN response by the inhibition of the dsDNA-sensing pathway. The full-length protein plays a role in virus replication in non-dividing cells, in the prevention of DCs maturation, and in the suppression of the IFN response by blocking the RNA-sensing pathway.

**Figure 4 ijms-23-13643-f004:**
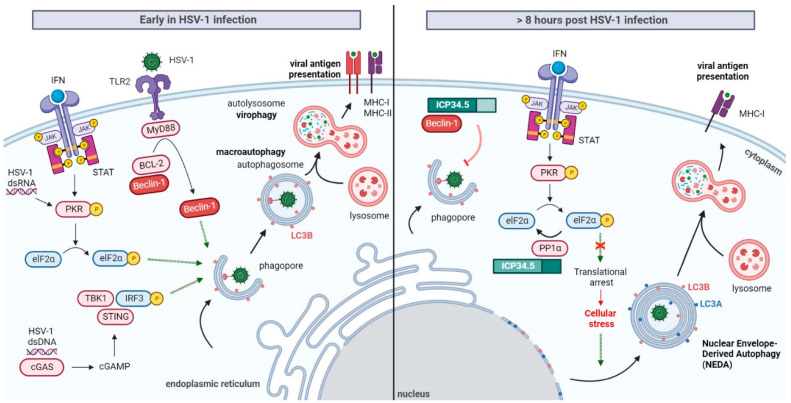
Autophagy modulation by ICP34.5 in HSV-1 infection. Early in HSV-1 infection, detection of the virus by host cell promotes the stimulation of macroautophagy. This pathway may act as a cellular defense mechanism involved in virophagy and the processing of viral antigen for MHC presentation. Autophagic flux can be induced in manner that is dependent on viral gene expression by the PKR/elF2α pathway. Autophagy may be enhanced through the recognition of PAMPs by the surface receptor TLR2. Activated TLR2 recruits the adaptor MyD88, which causes the dissociation of Beclin-1 from the BCL-2 inhibitory complex, resulting in the induction of autophagy. Finally, HSV-1 dsDNA may be recognized by the cytosolic dsDNA-sensing cGAS, which produces cGAMP to activate STING and TBK1. Activation of TBK1 promotes autophagy stimulation.

**Figure 5 ijms-23-13643-f005:**
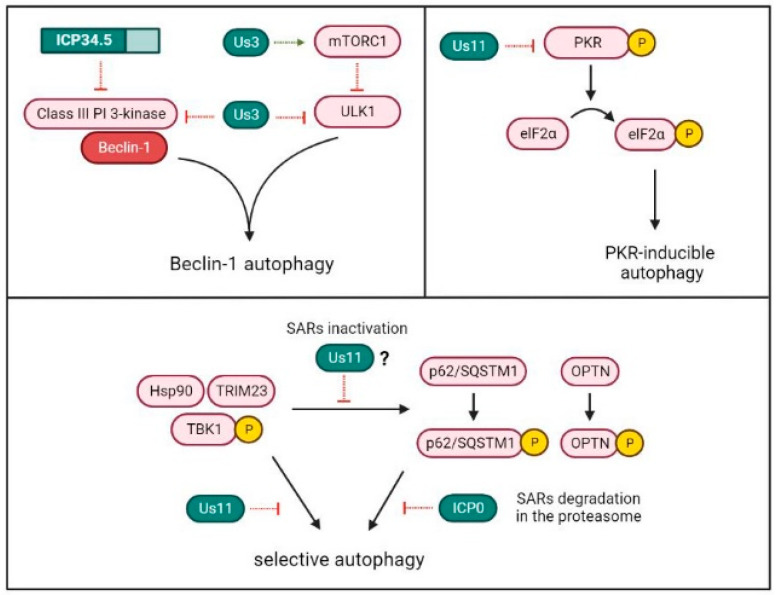
HSV-1 mechanisms for autophagy inhibition. HSV-1 proteins Us3 and ICP34.5 can inhibit macroautophagy by binding to Beclin-1, which is required for autophagosome formation. Us3 can also suppress the pathway by the activation of the negative regulator of autophagy mTORC1 or by the inactivation of the ULK1 complex, which is involved in phagophore initiation. The HSV-1 protein Us11 can prevent the induction of autophagy mediated by the PKR/elF2α pathway. Besides, Us11 inhibits virus-induced autophagy by the disassembly of the TRIM23-Hsp90-TBK1 complex. Finally, the HSV-1 protein ICP0 promotes the degradation in the proteasome of the selective autophagy receptors p62/SQSTM1 and OPTN, which are implied in the recruitment of HSV-1 in autophagosomal membranes. The TRIM23-TBK1 complex is involved in the phosphorylation and activation of these selective receptors. However, the inhibitory role of Us11 on the TRIM23-TBK1 complex has not been directly associated with a possible SARs inactivation.

## Data Availability

Not applicable.

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
