# Peer review of "Interplay between Autophagy and Herpes Simplex Virus Type 1: ICP34.5, One of the Main Actors"

_ijms, 2022, doi:10.3390/ijms232113643_

Round 1
Reviewer 1 Report
I find this review article to be very complete and an accurate review of HSV-1 and autophagy.
I find that they highlighted a important finding that HSV US3 may also antagonize autophagy in HSV-1 infected cells. The authors also correctly pointed out that the role of autophagy in HSV-1 infected cells depends greatly on the cell type.
One slight critical remark is that this review focuses solely on HSV-1 infection and does not mention even HSV-2 or other alpha herpesviruses.
Otherwise, I recommend publishing after spell check.
Author Response
First of all, I would like to thank the referee for the reviewing of the manuscript. We have attached the updated version of the review with the corresponding corrections.
Following your recommendation to include information about other alpha herpesviruses apart from HSV-1, we have added at the end of the review a short section about the relationship between HSV-2 and autophagy, focusing on the differences between ICP34.5 of HSV-1 and HSV-2. Finally, we have attended the request of minor spell check. Language changes made in the review are also indicated in green.

Reviewer 2 Report
In the presented review, Ripa et al. provide a very thorough and comprehensive picture relative to autophagy modulation upon infection with HSV-1 and the role of the ICP34.5 protein. The work is very well written and presented and just minor adjustments are required before final publication.
1) Authors could add a box or picture in the Introduction section, in which autphagy principles and mechanisms are summarized. This might help the reader with the numerous information provided in the introduction.
2) Analogously, a cartoon representation of HSV-1 infection might complete the review.
Author Response
First of all, I would like to thank the referee for the reviewing of the manuscript. We have attached the updated version of the review with the new figure (Figure 1).
Following your recommendation to include a picture about autophagy principles, we have made a new figure showing the different steps of autophagosome biogenesis and the main ATG proteins involved in this process. Regarding the cartoon representation of HSV-1 infection, we consider that incorporating a figure focused only on the viral cycle of HSV-1 might be a distraction from the central focus of the review. Because of this, instead of adding two different pictures, one for autophagosome biogenesis and another one for the HSV-1 replicative cycle, we have thought that it could be a good option to make a unique figure summarizing the main steps of macroautophagy and autophagosome formation, but also showing how HSV-1 interacts with autophagosomal membranes. We hope that this new figure helps to understand both the introduction of the review and the mechanism by which HSV-1 is degraded into autophagolysosomes.
